# Magnetically Reusable Fe$_3$O$_4$@NC@Pt Catalyst for Selective Reduction of Nitroarenes

**Jun Qiao** [1,2]**, Tian Wang** [2]**, Kai Zheng** [2]**, Enmu Zhou** [3]**, Chao Shen** [2] **, Aiquan Jia** [1] **and Qianfeng Zhang** [1,*]

1. School of Materials Science and Engineering, Institute of Molecular Engineering and Applied Chemistry, Anhui University of Technology, Ma'anshan 243002, China; qiaojun84@126.com (J.Q.); jaiquan@ahut.edu.cn (A.J.)
2. College of Biology and Environmental Engineering, Zhejiang Shuren University, Hangzhou 310015, China; workhard84@126.com (T.W.); zkai86@163.com (K.Z.); shenchaozju@163.com (C.S.)
3. College of Petroleum Chemical Industry, Changzhou University, Changzhou 213164, China; geormochou@outlook.com
* Correspondence: zhangqf@ahut.edu.cn; Tel.: +86-0555-2311059

**Abstract:** A novel reusable Fe$_3$O$_4$@NC@Pt heterogeneous catalyst was synthesized by immobilizing platinum on nitrogen-doped carbon magnetic nanostructures. It was characterized by infrared analysis (FT-IR), X-ray diffraction (XRD), transmission electron microscopy (TEM), and vibrating sample magnetometer (VSM). The catalytic efficiency of Fe$_3$O$_4$@NC@Pt was investigated by reduction of nitro aromatic compounds. The catalyst showed good catalytic activity, wide range of substrates, and good chemical selectivity, especially for the substrates of compounds containing halide and carbonyl groups. The magnetically catalyst can readily be reused up to ten cycles without loss of catalytic activity. Moreover, the key pharmaceutical intermediate *Lorlatini* can be facilely achieved through this strategy.

**Keywords:** reusable; magnetic; platinum; selective reduction; nitroarenes



## 1. Introduction

Anilines are one of the most important intermediates in fine chemicals, which can manufacture highly value-added chemicals such as natural products, pharmaceuticals, agrochemicals, surfactants, plastics, polymers, dyes, and pigments [1–5]. Traditionally, it can be synthesized by reduction of the corresponding nitroarenes using stoichiometric amounts of base metals like Fe/Zn and sulfides as the reducing reagents [6]. However, these reduction procedures may lead to vast amounts of solid wastes and hazardous by-products, causing serious environmental problems [7,8].

Catalytic reductions by heterogeneous transition metal catalysts have attracted much interest as a sustainable and efficient strategy because the catalysts can be easily separated and recovered from the reaction mixtures for reuse [9–11]. Magnetic heterogeneous catalysts have attracted remarkable attention by organic chemists due to their feasible removal and reusing by means of an external magnetic field [12–15]. A heterogeneous catalyst can be formed by loading active metals or molecular catalysts onto magnetic materials, which is not time-consuming and prevents losing of catalyst during the separation process. Supported noble metal catalysts like Pd, Au, Ag, Ru, and Rh have been widely investigated for the selective reduction of various substituted nitroarenes to the corresponding anilines due to their high activity [16–20]. However, these catalysts frequently generate some unwanted by-products, for example, dehalogenation of aromatic nitro compounds, which seriously lowers the product quality, limiting their large-scale industrial applications [21].

Biomass materials are widely used in industrial catalysis, carbon materials in particular, excellent catalytic activity was obtained by loading transition metals onto them [22–25], which are our longstanding interest [26–30]. Platinum has been demonstrated to exhibit much higher selectivity and dissociative capacity toward H2 in hydrogenation of

the nitroarenes. Meanwhile, its lower price and lower dehalogenation trend than palladium means it has great potential in the chemoselective hydrogenation of halogenated nitroarenes. In this paper, we find a facile approach to prepare a magnetically-reusable, nitrogen-doped carbon core-shell Fe$_3$O$_4$@NC@Pt catalyst for the selective reduction of nitroarenes, especially aromatic nitro compounds containing halogen and carbonyl substituent. The catalytic activity was optimized, and the substrate was expanded. The catalyst has also been used to synthesize *lorlatini*, the key pharmaceutical intermediates with high efficiency and cyclability.

## 2. Results and Discussion

From Scheme 1, the preparation process of Fe$_3$O$_4$@NC@Pt included three steps. Firstly, the Fe$_3$O$_4$ microspheres were prepared via a facile solvothermal reaction using FeCl$_3$·6H$_2$O as the raw material and ethylene glycol as the solvent. Secondly, a nitrogen-doped carbon layer was wrapped on the outside of Fe$_3$O$_4$ microspheres, the glucose served as a carbon source, and ethylenediamine (EDA) served as a nitrogen source. Finally, Pt(0) nanoparticles, obtained by reduction of H$_2$PtCl$_6$·6H$_2$O by sodium borohydride, were supported on Fe$_3$O$_4$@NC@Pt catalysts surface. To investigate the effects of nitrogen-doped carbon on the Fe$_3$O$_4$@NC@Pt catalytic activity of reduction of nitroarenes, Fe$_3$O$_4$@C@Pt was prepared in the same way mentioned above without adding ethylenediamine.

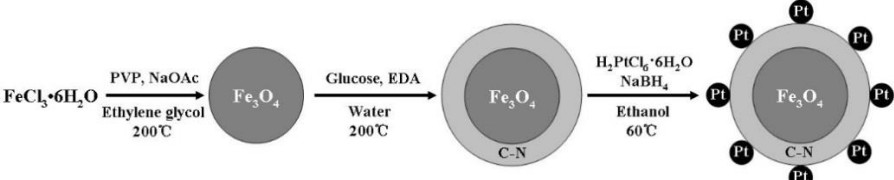

**Scheme 1.** Synthesis of Fe$_3$O$_4$@NC@Pt catalyst.

Figure 1 exhibits the FT-IR spectra of the Fe$_3$O$_4$, Fe$_3$O$_4$@NC, and Fe$_3$O$_4$@NC@Pt. In the spectrum (Figure 1a), the peaks were observed at around 583 cm$^{-1}$ (Fe-O stretching), 1626 cm$^{-1}$ (C=O bound stretching), and 1425 cm$^{-1}$ (pyrrole ring framework stretching), respectively. The characteristic peaks of Fe$_3$O$_4$@NC appeared at 579, 1689, and 3428 cm$^{-1}$ in Figure 1b, they were related to Fe-O, C=O band stretching, and the -OH stretching vibrations of carboxylic acids, respectively, indicating the surface functionalization on the surface of the Fe$_3$O$_4$@NC. It is very important for Pt immobilization with the help of these functional groups. As shown in Figure 1c, the absorption peaks of Fe$_3$O$_4$@NC@Pt at 581 and 1705 cm$^{-1}$ related to Fe$_3$O$_4$@NC. Meanwhile, it can be seen that the peak intensity of Fe-O, C=O, and -OH was obviously weaker compared with Fe$_3$O$_4$@NC. This may suggest that the interactions exist between platinum and Fe$_3$O$_4$@NC. A similar conclusion was reached in previous reports [31].

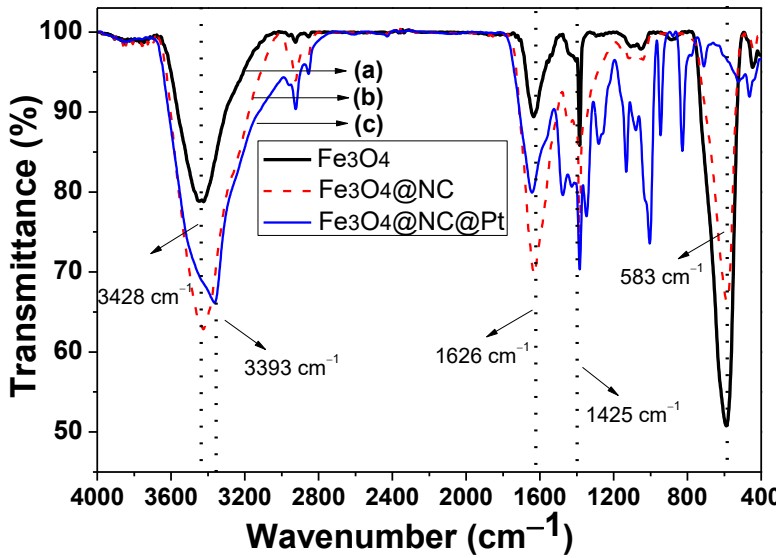

**Figure 1.** FT-IR spectra of (**a**) Fe$_3$O$_4$; (**b**) Fe$_3$O$_4$@NC; (**c**) Fe$_3$O$_4$@NC@Pt.

The catalyst composition and formulation were characterized by X-ray diffraction (XRD). In Figure 2a–c, the peaks at 2θ values of 18.2°, 30.1°, 35.4°, 37.1°, 43.1°, 53.4°, 53.4°, 56.9°, and 62.5° are well in agreement with the pure Fe$_3$O$_4$ (JCPDS 74-0748) [32,33]. In Figure 2c, the peaks at 2θ values of 40.0°, 46.5°, and 67.8° are attributed to the Pt (JCPDS 01-1194) [14]. All XRD patterns have no other impurities peaks, the sharp and strong diffraction peaks also confirm the well crystallization of the products. Meanwhile, in Figure 2b,c, the broad peak at 2θ = 21.6° is the characteristic reflection of amorphous nitrogen-doped carbon [34].

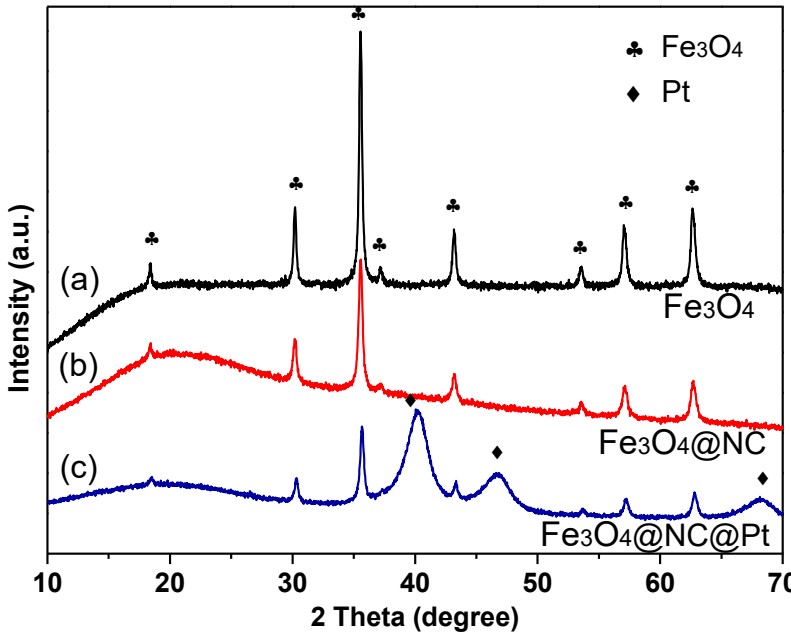

**Figure 2.** XRD patterns of (**a**) Fe$_3$O$_4$; (**b**) Fe$_3$O$_4$@NC; (**c**) Fe$_3$O$_4$@NC@Pt.

The specific surface area and pore structure of Fe$_3$O$_4$@NC@Pt were examined using low-temperature nitrogen adsorption–desorption measurements, the specific surface area was calculated by BET method [35], and the pore size distribution was calculated by Barrett–Joyner–Halenda (BJH) method [36]. From Figure 3, it can be seen that the isotherm of Fe$_3$O$_4$@NC@Pt is type IV compared with various type isotherms classified by the

IUPAC [37]. The isotherms showed a small hysteresis loop which is the characteristic of mesoporosity. The specific surface area is 5.8781 m$^2$/g and the total pore volume is 0.019498 cm$^3$/g, respectively. As shown in the BJH pore size distribution curves (Figure 3, inset), the catalyst pores are mainly in the range of 5–80 nm, the average diameter of the catalyst is 19.8 nm.

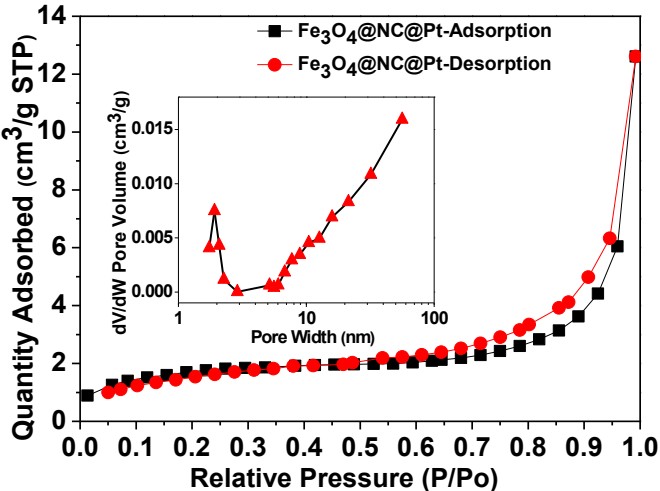

**Figure 3.** N$_2$ adsorption/desorption isotherms of Fe$_3$O$_4$@NC@Pt and BJH pore size distribution (inset).

Vibrating sample magnetometer (VSM) was used to test the magnetic properties of the Fe$_3$O$_4$@NC@Pt in fields ranging from +30 to −30 kOe at room temperature (Figure 4). The Fe$_3$O$_4$@NC@Pt shows ferromagnetic properties from the hysteresis loop, whose saturation magnetization value was 60.8 emu/g. With the help of an external magnetic field, the Fe$_3$O$_4$@NC@Pt could be quickly separated from the reaction system and reused for the next time.

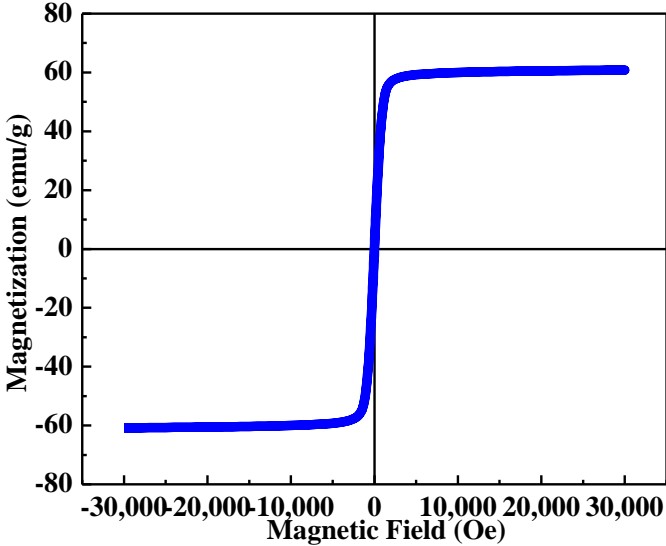

**Figure 4.** Vibrating Sample Magnetometer (VSM) data of Fe$_3$O$_4$ @NC@Pt.

Transmission Electron Microscopy (TEM) was used to further confirm the micromorphology and structure of materials. The prepared catalysts have obvious core structure (Fe$_3$O$_4$ Core), and small particles can be found on the surface of the core, the catalyst particles are uniform, and the size is ca. 3–4 μm (Figure 5a). In addition, the EDS mapping verified the presence of C, N, O, Fe, and Pt in Figure 5b and the dispersion and confinement

of Pt on Fe$_3$O$_4$@NC@Pt. The EDS mapping of Fe$_3$O$_4$@NC@Pt showed (Figure 5c) that the content of the platinum is 16.36% in semi-quantified.

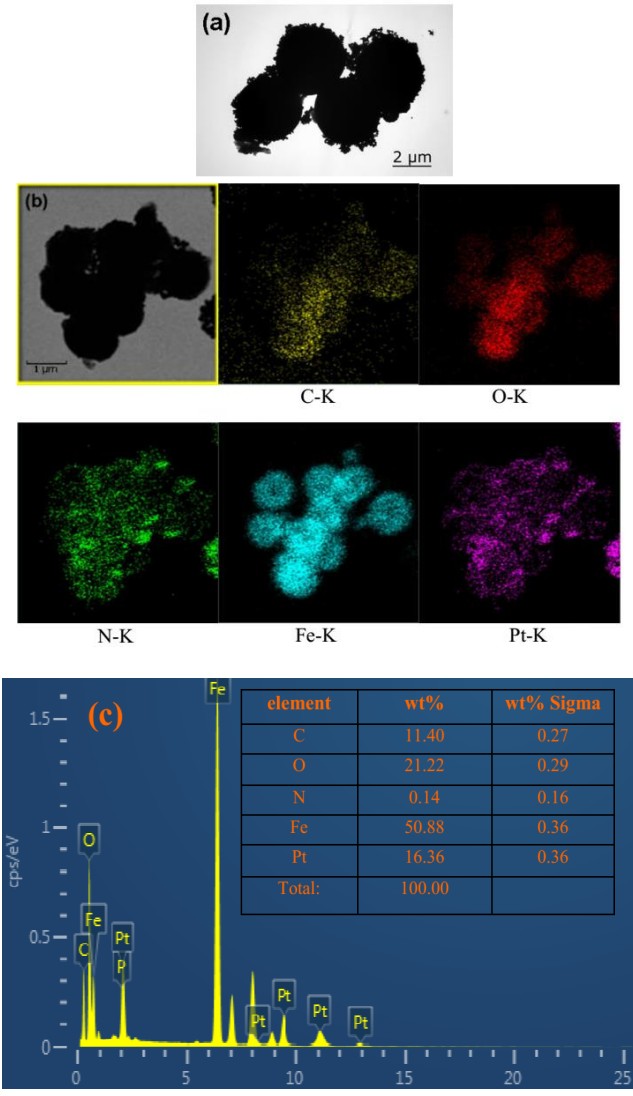

**Figure 5.** TEM image of Fe$_3$O$_4$@NC@Pt (**a**) and EDS elemental mappings of C, N, O, Fe, and Pt for Fe$_3$O$_4$@NC@Pt (**b**), EDS map sum spectrum pattern of Fe$_3$O$_4$@NC@Pt (**c**).

The magnetic catalyst Fe$_3$O$_4$@NC@Pt was evaluated in the reduction reaction. 1-chloro-4-nitrobenzene **1a** was used as the model compound to explore the catalytic activity of Fe$_3$O$_4$@NC@Pt. As we all know, the catalysts and reducing agents are two important factors for reduction reaction. For this purpose, a series of controlled experiments were studied, toluene was used as solvent and hydrazine hydrate was reducing agent, and the reaction temperature was 70 °C. The reduction reaction did not occur when Fe$_3$O$_4$@NC was used in the reaction, nevertheless, when replaced by Fe$_3$O$_4$@C@Pt, it showed much better catalytic activity but was less than Fe$_3$O$_4$@NC@Pt under the same conditions. Meanwhile, the reaction cannot occur without adding any catalyst (Table 1, entries 1–4). Therefore, two conclusions can be drawn: firstly, platinum (Pt) particles on the surface of the catalyst can effectively catalyze the reduction of nitro compounds to amine; secondly, N doped Fe$_3$O$_4$@NC@Pt catalyst is further improved in the catalytic activity, indicating N doping promotes the electron transfer on the catalyst surface during the reaction [38]. The reducing agent plays an important role in the reduction of aromatic nitro compounds, so the investigation of reducing agent is particularly important. First,

the effect of different reducing agents on the yield of the reaction was studied (Table 1, entries 4–7). It can be seen that hydrazine hydrate can reduce the 1-chloro-4-nitrobenzene very well, while hydrogen, sodium borohydride, and ammonium formate are not very effective, and hydrazine hydrate is also a commonly used reducing agent which is green, therefore, hydrazine hydrate was chosen as the optimized reducing agent. Furthermore, as a contrast, available comparative literature data obtained in similar conditions are listed in Table 1, entries 8–10, where it can be seen that these works are excellent.

**Table 1.** Optimization of catalyst and reducing agent [a].

| Entry | Catalyst | Reducing Agent | Temp (°C) | Time(h) | Yield [b] (%) | Ref. |
|---|---|---|---|---|---|---|
| 1 | - | hydrazine hydrate | 70 | 4 | - | this work |
| 2 | $Fe_3O_4$@NC | hydrazine hydrate | 70 | 4 | - | this work |
| 3 | $Fe_3O_4$@C@Pt | hydrazine hydrate | 70 | 4 | 70 | this work |
| 4 | $Fe_3O_4$@NC@Pt | hydrazine hydrate | 70 | 4 | 95 | this work |
| 5 | $Fe_3O_4$@NC@Pt | Hydrogen | 70 | 4 | 85 | this work |
| 6 | $Fe_3O_4$@NC@Pt | Sodium borohydride | 70 | 4 | 70 | this work |
| 7 | $Fe_3O_4$@NC@Pt | Ammonium formate | 70 | 4 | 30 | this work |
| 8 | Pt-Co/S-C | 2 bar $H_2$ | 40 | 2 | 99 | [39] |
| 9 | Pd–Pt–$Fe_3O_4$ | $NH_3BH_3$ | RT | 0.83 | 92 | [14] |
| 10 | Pt/RGO_EG | 1 bar $H_2$ | 40 | 2 | 96 | [25] |

[a] The reaction conditions: 1-Chloro-4-nitrobenzene **1a** (1 mmol), 4 mmol reducing agent, 0.5 mol% catalyst, and Toluene (3 mL), under air.
[b] Isolated yield.

As is known, the solvent also has an important effect on the reaction, and various solvents were used to track the reaction (Table 2, entries 1–8). Surprisingly, when using water as solvent, the yield of the product almost reached 99%, compared with the others for this reaction. The result was exciting because water as a solvent is green and nontoxic, which is more sustainable as a solvent for this reaction. As far as the effect of temperature on the reaction is concerned, interestingly, the reaction could also occur at room temperature, but in lower yields it can be seen (Table 2, entries 9–10). The reaction time was also one of the most important factors affecting the yield of the amine-based compounds. The results showed that with the decrease of reaction time, the yield of the product decreased, and the yield was only 65% for 2 h under the optimal conditions compared with 4 h (Table 2, entries 1 and 11). It can be concluded that the final optimal reaction conditions were confirmed: catalyst ($Fe_3O_4$@NC@Pt), reducing agent (hydrazine hydrate), solvent ($H_2O$), temperature (70 °C), and reaction time (4 h).

**Table 2.** Optimization of reaction conditions [a].

| Entry | Catalyst | Solvent | Temp(°C) | Time(h) | Yield [b] (%) |
|---|---|---|---|---|---|
| 1 | $Fe_3O_4$@NC@Pt | $H_2O$ | 70 | 4 | 99 |
| 2 | $Fe_3O_4$@NC@Pt | DMSO | 70 | 4 | 30 |
| 3 | $Fe_3O_4$@NC@Pt | Toluene | 70 | 4 | 95 |
| 4 | $Fe_3O_4$@NC@Pt | EtOH | 70 | 4 | 40 |
| 5 | $Fe_3O_4$@NC@Pt | Isopropanol | 70 | 4 | 50 |
| 6 | $Fe_3O_4$@NC@Pt | AcOEt | 70 | 4 | 10 |
| 7 | $Fe_3O_4$@NC@Pt | THF | 70 | 4 | 20 |
| 8 | $Fe_3O_4$@NC@Pt | Hexane | 70 | 4 | 50 |
| 9 | $Fe_3O_4$@NC@Pt | $H_2O$ | 50 | 4 | 70 |
| 10 | $Fe_3O_4$@NC@Pt | $H_2O$ | rt | 4 | 40 |
| 11 | $Fe_3O_4$@NC@Pt | $H_2O$ | 70 | 2 | 65 |
| 12 | Pt/CMK-3–HQ | EtOH | 80 | 1 | 99 [c] |

[a] the reaction conditions: 1-chloro-4-nitrobenzene **1a** (1 mmol), 4 mmol $N_2H_4 \cdot H_2O$, 0.5 mol% catalyst and solvent (3 mL) under air, all reaction temperature was 70 °C and the reaction time was 4 h, unless otherwise specified. [b] Isolated yield. [c] the work was done by the Xionggang Lu research group [21], as a comparison.

To demonstrate the general applicability, the selective hydrogenation of the $Fe_3O_4$@NC@Pt catalyst was further evaluated for 23 substituted nitroarenes in Table 3. Each substituted nitroarene could be transformed to corresponding aromatic amine in the presence of $Fe_3O_4$@NC@Pt catalyst, which showed excellent activity and selectivity. It also can be seen from Table 3 that the yields were all above 93%. From the experimental results, it can be found that the electronic effects of aryl nitro compounds with different substituents hardly affects the yield of the product. First, it can be found that all the halogenated nitroarenes compound could be favorably reduced to desired haloanilines almost without any by-product, however, 4-iodo nitrobenzene exhibited much lower reaction rates (Table 3, **2a–2f**), the yield of 4-iodine aniline was lower than the others halogenated nitroarenes, as we know it is a tendency to dehalogenate for iodoaromatic nitro compounds more easily during reduction. Compounds with carbonyl groups were also reduced when the nitro groups was reduced in the process (Table 3, **2g–2j**). After 8 h of reduction, 4-nitrobenzaldehyde was reduced to 4-amino benzaldehyde (yield 99%), when the aldehyde group was replaced by other reductive groups, such as -$COCH_3$ and -COOH, the yield of the product could also be obtained according to the same strategy. In addition, other nitroarene compound containing sulfo, alkyl, alkoxy, amino, and biphenyl, could be reduced to corresponding anilines in good yields (Table 3, **2k–2v**). Finally, a heterocyclic aromatic nitro compound was selected as a substrate, which could also be reduced to amino, and the yield was 95% (Table 3, **2w**). All [1]H NMR spectrum of products (**2a–2w**) (See supporting information Figures S1–S23).

**Table 3.** Fe$_3$O$_4$@NC@Pt catalyzed hydrogenation of different nitroarenes [a].

|  |  |  |  |  |
|---|---|---|---|---|
| **2a**, 99% | **2b**, 95% [b] | **2c**, 93% [b] | **2d**, 98% [b] | **2e**, 99% [b] |
| **2f**, 99% | **2g**, 97% [b] | **2h**, 98% [b] | **2i**, 95% [c] | **2j**, 94% [b] |
| **2k**, 98% | **2l** 97% | **2m**, 95% [b] | **2n**, 98% | **2o**, 99% |
| **2p**, 99% | **2q**, 98% | **2r**, 94% [b] | **2s**, 99% | **2t**, 98% |
| **2u**, 96% | | **2v**, 95% [b] | | **2w**, 95% [c] |

[a] The reaction conditions: nitroarenes 1a (1 mmol), N$_2$H$_4$·H$_2$O (4 mmol), 0.5 mol% catalysts, 4 h and H$_2$O (3 mL) under air, isolated yield; b 8 h; c 16 h.

For industrial applications of catalytic systems, the Fe$_3$O$_4$@NC@Pt catalytic hydrogenation strategy was used to obtain the important pharmaceutical intermediates of commercial drug Lorlatini (Scheme 2). Compound **4** had great significance, which can be served as precursors for the synthesis of antitumor drugs *Lorlatini*. Typically, 4 mmol compounds **3** was input to 6 mL distilled water and then 0.080 g ultrasonically dispersed Fe$_3$O$_4$@NC@Pt catalyst in water (8 mL) was introduced to this solution. Next, 10 mmol N$_2$H$_4$·H$_2$O was added, and the mixture was stirred in 70 °C for 4 h. The reaction system was detected by TLC. Conversion from compound **3** to **4** with excellent results in the presence of Fe$_3$O$_4$@NC@Pt catalyst in water at 70 °C for 4 h. The target product **4** was achieved with high isolated yield (up to 99% yield). After the reaction, Pt in the reaction solvent was determined by ICP-OES, however, no platinum contamination was detected (Pt content < 10 ppm). The $^1$H NMR spectrum of products **4** (See supporting information Figure S24).

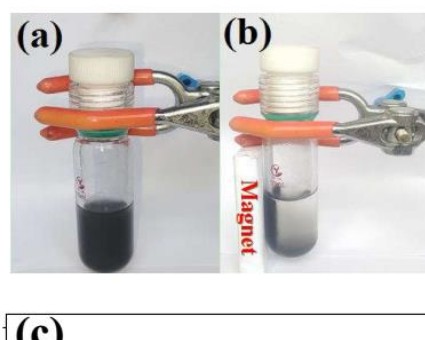

**Scheme 2.** Application of the catalyst in the synthesis of intermediate of commercial drug *Lorlatini*.

For heterogeneous catalysts, the stability, ease of separation, and reusability are the most important features. First, hot leaching experiment was conducted when reacted for 1 h, the catalyst $Fe_3O_4$@NC@Pt was recovered with a magnet, and then the system without a catalyst continued to react at 70 °C for 3 h. 1-chloro-4-nitrobenzene conversion maintained the same after removal of the catalyst. This further confirmed that Pt particles were immobilized onto the catalyst surface and not leached into the reaction solution, the dispersion state photographs of the magnetically separable $Fe_3O_4$@NC@Pt (Figure 6a) and magnetic separation of $Fe_3O_4$@NC@Pt after reaction (Figure 6b). The recyclability of $Fe_3O_4$@NC@Pt was tested for the hydrogenation of 1-chloro-4-nitrobenzene in Figure 6c. In cycle performance test, the yield of 4-chloroaniline was still good for the sixth run. However, the catalytic activity was slightly reduced, and the yield of aniline remained at 92% after ten cycles. All these results suggested that the $Fe_3O_4$@NC@Pt catalyst could maintain activity without a significant efficiency reduction.

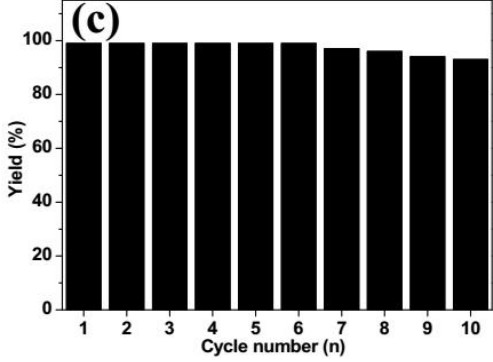

**Figure 6.** Photographs of the magnetically separable $Fe_3O_4$@NC@Pt (**a**) dispersion state of $Fe_3O_4$@NC@Pt; (**b**) magnetic separation of $Fe_3O_4$ @NC@Pt after reaction; (**c**) recyclability of the $Fe_3O_4$@NC@Pt catalyst.

## 3. Experimental Materials

The reagent was commercially available Ferric chloride hexahydrate ($FeCl_3 \cdot 6H_2O$), Glucose, Sodium acetateanhydrous, ethylene glycol, Vinylpyrrolidinone polymer (PVP), and ethylenediamine (EDA), which were purchased from Shanghai Aladdin Reagent Co. Ltd (Shanghai, China). Chloroplatinic acid hexahydrate ($H_2PtCl_6 \cdot 6H_2O$, $\geq 37.50\%$) was purchased from J&K Scientific Ltd (Beijing, China). All reagents are analytically pure and used without further purification.

### 3.1. Characterization

Fourier-transform infrared (FTIR) spectra were recorded with a Bruker Tensor 27 using KBr pellets(4000–400 cm$^{-1}$). Transmission electron microscopy (TEM) images were performed on a JEOL 2100 TEM microscope operating at 200 kV. The X-ray diffraction (XRD) ware tested on D/max 2500 PC X-ray diffraction (XRD, Rigaku, Tokyo, Japan) with Cu-K$\alpha$ radiation ($\lambda$= 0.154 nm) operated at 40 kV and 40 mA. The N$_2$ adsorption and desorption isotherms were recorded on a Micromeritics ASAP 2020 sorptometer at −196 °C, and the samples were degassed for 10 h at 180 °C before analysis. The magnetic properties of the catalysts were tested on a vibrating sample magnetometer (VSM) at room temperature from +30 to −30 kOe. $^1$H and $^{13}$C NMR spectra were carried out with a Bruker Advance 400 spectrometer by using TMS as the internal standard and DMSO-$d_6$ or CDCl$_3$ as solvents. The Pt content in the catalyst was carried out with a Perkin-Elmer Optima 2100 DV.

### 3.2. Preparation of Fe₃O₄ Microspheres

Fe$_3$O$_4$ microspheres were obtained according to a previous report [40]. In a typical experiment, 3 g FeCl$_3$·6H$_2$O, 4 g NaAc, and 2 g PVP were dispersed in 60 mL ethylene glycol under magnetic stirring. Next, the mixture was transferred to a Teflon-lined autoclave and maintained at 200 °C for 12 h. After that, the desired product was separated using an external magnet and washed with water and ethanol for several times, respectively. Finally, the black Fe$_3$O$_4$ microspheres were dried in a vacuum oven at 60 °C for 24 h.

### 3.3. Preparation of Fe₃O₄@C Microspheres

Fe$_3$O$_4$@C was fabricated by a simple carbonization of glucose on Fe$_3$O$_4$ surface under hydrothermal conditions. 200 mg Fe$_3$O$_4$ microspheres were dispersed in 10 mL water of glucose (3.2 g) aqueous solution and ultrasonicated for 0.5 h. After that, they were added into a 100mL Teflon lined autoclave heated at 180 °C for 10 h. After cooling to room temperature, the final Fe$_3$O$_4$@C microspheres were obtained by an external magnet and washed with ethanol, followed by water. Finally, the product with black color was dried in a vacuum oven for 24 h.

### 3.4. Preparation of Fe₃O₄@NC Microspheres

Fe$_3$O$_4$@NC was synthesized according to the previous report described elsewhere [41]. 100 mg Fe$_3$O$_4$ microspheres were dispersed into 10 mL water of ethylenediamine (EDA) (0.2 mL) and 1.6 g glucose solution, then ultrasonicated for 0.5 h. Subsequently, the mixture was transferred into a Teflon-lined autoclave and treated at 180 °C for 10 h. After cooling to room temperature, the obtained black product Fe$_3$O$_4$@NC microspheres were washed with ethanol and water, respectively. Lastly, they were dried in a vacuum oven for 24 h.

### 3.5. Preparation of the Fe₃O₄@C@Pt and Fe₃O₄@NC@Pt Catalyst

The Fe$_3$O$_4$@C@Pt and Fe$_3$O$_4$@NC@Pt catalyst were prepared according to the previous report, with some modifications [42]. In a typical procedure, take the Fe$_3$O$_4$@NC preparation process for example, 400 mg Fe$_3$O$_4$@NC were dispersed in 40 mL ethanol by ultrasonic treatment for 0.5 h. Next, 3 mL H$_2$PtCl$_6$·6H$_2$O (35 mg) of ethanol solution were added into Fe$_3$O$_4$@NC suspension solution and continuously ultrasonicated for 1 h, the last, 8 mL sodium borohydride (1 g) of ethanol solution was dropped into the above mixture with vigorous stirring under 60 °C. After 2 h of reduction, the products were separated and washed several times with water. The products were dried in a vacuum oven to obtain Fe$_3$O$_4$@NC@Pt. The same prepared process was undertaken for Fe$_3$O$_4$@C@Pt.

### 3.6. Procedure for the Selective Hydrogenation of Nitroarenes Reactions

1 mmol nitro compounds was input to 2 mL distilled water, and then 0.020 g ultrasonically dispersed Fe$_3$O$_4$@NC@Pt catalyst in water (2 mL) was introduced to this solution. Next, 4 mmol N$_2$H$_4$·H$_2$O was added, and the mixture was stirred in 70 °C. The reaction

system was detected by TLC. After that, the catalyst was removed with external magnetic field and washed several times with ethanol and used after drying in subsequent reactions, then the residual solvent was evaporated under vacuum to obtain the pure amines. The conversions were determined by the gas chromatography (GC) analysis. All the synthesized amines were characterized by comparison of NMR spectral data with the reported values in literatures.

### 3.7. Procedure for Catalyst Reused

1.0 mmol 1-chloro-4-nitrobenzene, 4.0 mmol $N_2H_4 \cdot H_2O$, and $Fe_3O_4$@NC@Pt were mixed in 3 mL $H_2O$. The mixture was stirred at 70 °C. After the reaction, the catalyst was separated by an external magnet and washed with water for three times (2 mL Water/time) and ethanol for three times (2 mL ethanol/time), then dried in vacuum and directly used in the next run.

## 4. Conclusions

A novel magnetically recyclable $Fe_3O_4$@NC@Pt catalyst was prepared. The results show that the catalyst has a good catalytic activity and especially high selectivity for the hydrogenation of nitroarenes with halogen and carbonyl groups under mild reaction conditions. These magnetic catalysts could readily be reused at least ten times without significant decrease of catalytic activity. Moreover, the key pharmaceutical intermediates *Lorlatini* could be easily synthesized using this catalyst. Therefore, this new catalyst system should be a very useful, sustainable, and environmentally friendly solution in industrial applications.

**Supplementary Materials:** The following are available online at https://www.mdpi.com/article/10.3390/catal11101219/s1, Figure S1: [1]H NMR spectrum of **2a**, recorded in Chloroform-*d* at 25 °C; Figure S2: [1]H NMR spectrum of **2b**, recorded in Chloroform-*d* at 25 °C; Figure S3: [1]H NMR spectrum of **2c**, recorded in DMSO-*$d_6$* at 25 °C; Figure S4: [1]H NMR spectrum of **2d**, recorded in Chloroform-*d* at 25 °C; Figure S5: [1]H NMR spectrum of **2e**, recorded in Chloroform-*d* at 25 °C; Figure S6: [1]H NMR spectrum of **2f**, recorded in DMSO-*$d_6$* at 25 °C; Figure S7: [1]H NMR spectrum of **2g**, recorded in Chloroform-*d* at 25 °C; Figure S8: [1]H NMR spectrum of **2h**, recorded in Chloroform-*d* at 25 °C; Figure S9: [1]H NMR spectrum of **2i**, recorded in Chloroform-*d* at 25 °C; Figure S10: [1]H NMR spectrum of **2j**, recorded in Chloroform-*d* at 25 °C; Figure S11: [1]H NMR spectrum of **2k**, recorded in Chloroform-*d* at 25 °C; Figure S12: [1]H NMR spectrum of **2l**, recorded in Chloroform-*d* at 25 °C; Figure S13: [1]H NMR spectrum of **2m**, recorded in Chloroform-*d* at 25 °C; Figure S14: [1]H NMR spectrum of **2n**, recorded in Chloroform-*d* at 25 °C; Figure S15: [1]H NMR spectrum of **2o**, recorded in Chloroform-*d* at 25 °C; Figure S16: [1]H NMR spectrum of **2p**, recorded in Chloroform-*d* at 25 °C; Figure S17: [1]H NMR spectrum of **2q**, recorded in Chloroform-*d* at 25 °C; Figure S18: [1]H NMR spectrum of **2r**, recorded in DMSO-*$d_6$* at 25 °C; Figure S19: [1]H NMR spectrum of **2s**, recorded in Chloroform-*d* at 25 °C; Figure S20: [1]H NMR spectrum of **2t**, recorded in DMSO-*$d_6$* at 25 °C; Figure S21: [1]H NMR spectrum of **2u**, recorded in Chloroform-*d* at 25 °C; Figure S22: [1]H NMR spectrum of **2v**, recorded in Chloroform-*d* at 25 °C; Figure S23: [1]H NMR spectrum of **2w**, recorded in Chloroform-*d* at 25 °C; Figure S24: [1]H NMR spectrum of **4**, recorded in DMSO-*$d_6$* at 25 °C.

**Author Contributions:** The experimental work was conceived and designed by J.Q. and Q.Z.; T.W., K.Z., and E.Z. performed the experiments; J.Q., C.S. and A.J. analyzed the data; J.Q. and Q.Z. drafted the paper. All authors have read and agreed to the published version of the manuscript.

**Funding:** This research was funded by Zhejiang Shuren University Basic Scientific Research Special Funds, grant number 2020XZ011 and 2021 Scientific and Technological Innovation Activity Plan for College Students in Zhejiang Province (New Talent Plan), grant number 2021R421009.

**Data Availability Statement:** There is no additional data available.

**Conflicts of Interest:** The authors declare no conflict of interest.

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
