# Peer review of "Magnetically Reusable Fe3O4@NC@Pt Catalyst for Selective Reduction of Nitroarenes"

_catalysts, doi:10.3390/catal11101219_

Round 1

Reviewer 1 Report

The Authors provide a brief introduction to the subject matter, present the aim of the work but, in Introduction, there is a lack of accurate references to earlier research carried out in the discussed thematic scope. In the next chapter, Results and Discussion, there is a lot to improve and to supplement. In general, the manuscript requires thorough revision in order to consider the next step of the proceeding and I suggest, in particular, an implementation of the following amendments:

  • The language of the manuscript must be thoroughly improved to meet standards of the language of scientific article.
  • A lack of detailed descriptions of the measurement methodology used in instrumental methods applied in the studies. A more comprehensive descriptions is needed here.
  • The physicochemical characteristic of the catalysts is superficial. More specific and detailed confirmation of composition, formulation and structure, i.e. identity of the catalysts is required.
    At the very beginning of chapter 2, it talks about catalysts characterization using, among others, ICP-OES technique. Nevertheless, there are no presented results and data obtained with the use of this technique. There is no discussion on the mentioned results which could verify assumption about the catalysts form.
  • The discussion of the FT-IR studies (page 2) is imprecise, even sloppy, and needs to be improved (e.g. I. much better justify the presence of -COOH group, II. what is meant by: “the -COOH bond”? – this is a mistake, III. band at 3126 cm-1 may concern O-H stretching vibrations in carboxylic acids, etc. – is much to clarify and explain).
  • The Figure 1 – illegible. The location of specific absorption bands have to be indicated, pointed out in the Figure.
    I suggest to insert the scale on the Y-axis to make visible an intensity of given bands and I suggest to appropriately enlarge the Figure.
  • A little short on details about the reaction contained in the Scheme 2. More detailed description of this process needs to be introduced, especially in terms of its carrying out by the Authors in their studies.
  • The recyclability of tested catalysts – little data about this matter were presented, concern only single series of the reaction with one type of substrate. The discussion on this issue is very limited and needs to be enhanced.
    In order to determine the real recyclability of the catalyst and its ability to retain the catalytic activity this part of research should be extended.
  • An error in the description of reducing agents in the discussion on page 5 – should be “entries 4-7” not “entries 1-4” (with reference to Table 1).
  • The optimization included only one reaction (with one specific substrate). It is worth to extend the optimization to other reactions (with different substrates) to perform comprehensive optimization of the whole process.

Reviewer 2 Report

This is a good article that might be suitable for publication after taking into account the following suggestion, which will certainly improve this manuscript.

  1. Line 2.  More details are needed about nanocomposites, especially structure and dimensions
  2. Please, give more motivational explanations why Fe3O4?
  3. Although the abstract refers to nanocomposite, there is no such word in the introduction.
  4. Why Fe3O4 and not Fe2O3?
  5. For a wider range of readers, more information on preparation methods and applications would be desirable. Quite a lot of old references, which are of course important and interesting. However, there are many new and promising uses. See for examples, some of them published this year in MDPI journals,

Serga, V.et al . Impact of Gadolinium on the Structure and Magnetic Properties of Nanocrystalline Powders of Iron Oxides Produced by the Extraction-Pyrolytic Method. Materials 202013, 4147.

Li, Y.; Wang, Z.; Liu, R. Superparamagnetic α-Fe2O3/Fe3O4 Heterogeneous Nanoparticles with Enhanced Biocompatibility. Nanomaterials 202111, 834.       

  1. Results and Discussion. Data , presented in Fig.1 and in 3rd  paragraph  it is better also to present in the Table with all supporting references, concerning  IR band assignments.
  2. Page 3. For BET method, please add the supporting reference. The same is for BJH.
  3. 2 description. Remove the extra dot before BJH.
  4. Is it possible to add in Table 1 and 2, available comparative literature data obtained in similar conditions?
  5. Page 3. “classified by IUPAC” needs proper reference.
  6. Page 9. “flleting” ?

Round 2

Reviewer 1 Report

I thank the Authors for their response and commentary. The Authors made some improvements, quite a few improvements they still could make to enhance the manuscript. I’d like to mark two amendments which I consider certainly necessary to be done.

  • The language should be further improved. I strongly suggest consultation with native speaker or professional editing service support.
  • The scale on the Y-axis in the Figure 1 is most likely incorrect. It’s good that Authors put it in the Figure but the actual scale must be placed there.
    I suggest:
    A) the scale starts from 100% transmittance (0% absorbance) for each of spectrum separately (spectra placed one below the other)
    or
    B) a joint scale with one 100% transmittance point and all spectra overlapped.
